# Phylogenomic Analyses of the Tenthredinoidea Support the Familial Rank of Athaliidae (Insecta, Tenthredinoidea)

**DOI:** 10.3390/insects13100858

**Published:** 2022-09-21

**Authors:** Gengyun Niu, Mahir Budak, Ertan Mahir Korkmaz, Özgül Doğan, André Nel, Siying Wan, Chenyang Cai, Corentin Jouault, Min Li, Meicai Wei

**Affiliations:** 1Laboratory of Insect Systematics and Evolutionary Biology, College of Life Sciences, Jiangxi Normal University, Nanchang 330022, China; 2Department of Molecular Biology and Genetics, Faculty of Science, Sivas Cumhuriyet University, Sivas 58140, Turkey; 3Institut de Systématique, Évolution, Biodiversité (ISYEB) Muséum National d’Histoire Naturelle, CNRS, Sorbonne Université, EPHE, Université des Antilles, CP50, 57 rue Cuvier, 75005 Paris, France; 4State Key Laboratory of Palaeobiology and Stratigraphy, Nanjing Institute of Geology and Palaeontology, Centre for Excellence in Life and Paleoenvironment, Chinese Academy of Sciences, Nanjing 210008, China; 5Univ. Rennes, CNRS, Géosciences Rennes, UMR 6118, F-35000 Rennes, France; 6CNRS, Institut des Sciences de l’Évolution de Montpellier, UMR 5554, 34090 Montpellier, France

**Keywords:** Tenthredinidae, phylogeny, rare genomic changes, gene rearrangement, RNA secondary structure

## Abstract

**Simple Summary:**

In the current era of data explosion, the use of genetic information is increasingly being applied across numerous biological questions. One application has been to develop more robust evolutionary frameworks. Such well-resolved phylogenetic relationships are currently lacking from many of the basal branches of diversity-rich taxa. This is most pronounced at the base of the thentredinoid, especially *Athalia*. This study reviews earlier comparative morphological studies and advances in phylogenetic studies based on morphological characters and short sequence fragments, using both mitochondrial and nuclear genetic sequences as well as genomic structural evidence to define the family Athaliidae in several dimensions and clarify its phylogenetic position. As a result, Athaliidae no longer belongs to the Tenthredinidae but is independent and distant from it. This study clarifies a hurdle to solve the scientific problem of hymenopteran evolution.

**Abstract:**

The systematic status of the genus *Athalia* and related genera is a perennial controversy in sawfly taxonomy. Several authors have hypothesized that the placement of *Athalia* within the Tenthredinidae is artificial, but no studies have focused on this topic. If the hypothesis that *Athalia* does not belong to Tenthredinidae can be supported, the taxonomic framework of Tenthredinoidea needs revision. We present a comprehensive phylogenomic study of Tenthredinoidae, focusing on the positions of *Athalia* and related genera by sampling 80 representatives mainly of the Tenthredinoidea, including Heptamelinae and Blasticotomidae. Our phylogenetic reconstructions based on nuclear genes and mitochondrial (mt) sequences support *Athalia* and related genera as a distinct clade sister to Tenthredinidae + (Cimbicidae + Diprionidae). A comparison of symphytan mitochondrial genomes reveals an innovative gene rearrangement pattern in Athaliidae, in which *Dentathalia* demonstrates a more ancestral pattern than *Athalia* and *Hypsathalia*. The lineage specificity of mt rRNA secondary structures also provides sufficient support to consider Athaliidae as a separate family. In summary, the phylogeny and genomic structural changes unanimously support the taxonomic treatment of Athaliidae as a family and the re-establishment of *Dentathalia* as a valid genus.

## 1. Introduction

Our current knowledge of the evolutionary relationships among insect orders is extensive and supported by numerous large-scale phylogenomic analyses [1,2,3,4,5,6,7,8,9]. However, inconsistencies persist regarding the phylogenetic relationships between a few groups (e.g., Psocoptera, Antliophora + Siphonaptera) and have challenged the perfect picture of insect evolution [1,10,11,12]. It is widely accepted that further advances in insect phylogenetics cannot rely only on deeper and broader genome and taxon sequencing. Improved modeling of the evolutionary process and a perpetually nuanced view are fundamental to clarifying and reducing the number of paraphyletic groups [13,14].

Genome sequences are the preferred data for phylogenetic relationship reconstruction because they contain abundant informative sites. Nevertheless, dissimilarity and inconsistency between data always exist. In contrast, rare genomic changes (RGCs) provide concise evidence [15]. The use of mitochondrial (mt) genomes, which are compact with nearly invariant gene contents, potentially offers lower error rates and a high potential to resolve deep nodes [16]. Even if gene orders (GOs) can be modeled as signed permutations, they cannot be treated directly as character matrices [17]. To extract phylogenetic information from GOs, specialized computational methods and tools have been developed. Among the model-based frameworks, a set of rearrangement operations known as the rearrangement model, pioneered by Watterson et al. [18], are used to measure pairwise distances. Then, it has been found that the tandem duplication random loss (TDRL) operation is more severe in rearranging GOs than other types of rearrangement operations in metazoans [19]. Several software tools that regard TDRL rearrangements were then developed for the reconstruction of evolutionary scenarios [20]. Notably, tandem duplication nonrandom loss mutations also occur [21], but they are not yet included in the abovementioned models. In addition to the model-based framework, the extraction of phylogenetic information directly from GOs without making assumptions about rearrangement operations is also useful. The results support one hypothesis among a set of competing hypotheses favored by different data [22]. Interestingly, the Hymenoptera are among the orders of insects with the highest variation in GOs [23]. The rearrangements occur in various taxa, with all three types of genes being variable [24]. As the sampling coverage increases, a large number of shared derived features are discovered. Many of these GOs provide additional support for monophyly [25] or for defining the boundaries of higher taxa [26]. These newly discovered mitochondrial genomes fill discontinuities and restore the continuous spectrum of evolutionary scenarios, calling for more active accumulation of data and a modification of the model.

In addition to GOs, other traits may be associated with strong selection, such as innovation in RNA genes, which are also considered to be useful in a phylogenetic context. Systematic studies of the evolution of non-coding RNA secondary structure have highlighted the importance of branch-specific structural insertion/deletion domains, but it appears that data accumulation was not sufficient to explore issues related to sequence signatures [27]. Moreover, the predicted secondary structures remain limited to a small number of taxa, thus preventing effective large-scale comparative studies. However, given the vast amount of data available because of next-generation sequencing (NGS) technology, the time has come to define the characteristics of predictive structures for application in phylogenetic studies.

Among the basal taxa of Hymenoptera, one of the most problematic paraphyletic groups is the Tenthredinidae (Figure 1). To clarify this issue, the position of the enigmatic genus *Athalia* Leach, 1817 and related genera need to be addressed.

### 1.1. Athalia and Its Relatives

*Athalia* is one of the twenty oldest described genera of Tenthredinidae andoccurs in the Eurasian and African regions [35].

*Hennedyia* Cameron, 1891 was described by Cameron [36]. The genus includes only one species and is found in southern Europe [37].

Benson [38] created the subgenus *Dentathalia* Benson, 1931, to accommodate the particular toothed claw of the species *Athalia scutellariae* Cameron, 1880 under the genus *Athalia*. However, later, *Dentathalia* was synonymized with *Athalia* by Benson [39].

*Hennedyella* Forsius, 1935 was described from Burma (=Myanmar) by Forsius [40]. The genus includes two species and occurs in Myanmar, Thailand [41], and the eastern Himalayas [42].

Benson [39] erected *Hypsathalia* Benson, 1962, for *Athalia przevalskyi* Jakovlev, 1887. The genus includes one known species and is distributed in Nepal, North India, and Tibet (China).

### 1.2. An Overview of the Systematic Position of Athalia and Its Relatives

The systematic positions of *Athalia* and its relatives, *Hypsathalia*, *Hennedyia*, and *Hennedyella*, are among the significant unresolved problems in the phylogeny of early-diverging lineages of Hymenoptera. Leach [35] classified Tenthredinidae *s. lat.* into nine stripes and placed the genus *Athalia* in stripe six with *Messa* Leach, 1817, *Selandria* Leach, 1817, and *Fenusa* Leach, 1817.

Hartig [43] divided Tenthredinidae *s. lat.* into four groups (Cimbicides, Hylotomides, Tenthredinides, and Lydides) and stated specifically that *Athalia* forms a “connecting link” between the Hylotomides and Tenthredinides.

Kirby [44] proposed the subfamily Athaliina, encompassing *Athalia*, and placed it within the family Tenthredinidae. In his system, Kirby divided the clades composing the “Symphyta” into two families: Tenthredinidae and Siricidae.

Cameron [45] recognized four families under Sessiliventris (=“Symphyta”). The family Tenthredinidae was classified into eight subfamilies, and *Athalia* was placed into the subfamily Tenthredina. Cameron pointed out that *Athalia* is a distinct and ‘ancient’ genus within the family.

Ashmead [46] divided the suborder Phytophaga into two series, Xylophaga and Phyllophaga. The genera *Athalia* and *Hennedyia* were placed into the Selandriinae of Selandriidae.

Konow [47] regarded Phyllophaga, except for Xyelidae and Lydidae, as one family: the Tenthredinidae. He subdivided the family Tenthredinidae into four subfamilies and placed *Athalia* into the tribe Selandriades (nearly equivalent to the current Selandriinae, Allantinae, and Cladiuchini) of Tenthredinini, and *Hennedyia* was placed into a heterogeneous tribe (Hoplocampides) with six other genera: *Phyllotoma* Fallen, 1829, *Poppia* Konow, 1904, *Heptamelus* Haliday, 1855, *Anapeptamena* Konow, 1898, *Eriocampoides* Konow, 1890, and *Hoplocampa* Hartig, 1837.

Rohwer [48] defined modern concepts of the family Tenthredinidae and the superfamily Tenthredinoidea. He further divided the family Tenthredinidae into eleven subfamilies, including Athaliinae, which lies between the Messinae (=Fenusinae) and the Empriinae (=Allantinae).

Benson [49] classified the Tenthredinidae into seven subfamilies, and the tribe Athaliini solely composed of *Athalia* was placed into the Emphytinae (=Allantinae). Note that the genera *Hennedyia* and *Hennedyella* were omitted from his system. Takeuchi [50] also placed the tribe Athaliini into Allantinae. However, Benson [51] changed his system and placed Athaliini into a broadly defined Blennocampinae (including Empriini, Allantini, Caliroini, Blennocampini, and Fenusini). Benson [39] revised the global genera and species of Athaliini. The three monotypic genera *Hypsathalia*, *Hennedyia,* and *Hennedyella* were placed into the Athaliini.

Lorenz and Kraus [52] studied the larval morphology of sawflies and divided the Tenthredinidae into six subfamilies (Dolerinae, Tenthredininae, Selandriinae, Blennocampinae, Nematinae, and Heterarthrinae). They placed the Athaliini into the comprehensive Blennocampinae, which corresponds now to the Allantinae, Blennocampinae, Fenusinini, Caliroini, Eriocampini, Athaliini, and *Mesoneura* Hartig, 1837.

Zhelochovtsev [53] divided the Tenthredinidae into two subfamilies (Tenthredininae and Nematinae), and the Athaliini were placed into the Tenthredininae.

Abe and Smith [54] divided the Tenthredinidae into eight subfamilies but without tribal arrangement. The genera *Athalia*, *Hypsathalia*, *Hennedyia,* and *Hennedyella* were placed into the Allantinae.

Wei [28] reconstructed the phylogeny of the Tenthredinidae and proposed a new system for the family. He divided the family Tenthredinidae into six families and 19 subfamilies (Figure 1a). The Athaliinae were placed into the Blennocampidae as the sister group of Lycaotinae. Wei and Nie [55] published a generic list of global Tenthredinoidea *s. str.* based on the data of Wei [28], first excluded the genus *Athalia* and its relatives from Allantinae and placed them as a subfamily, Athaliinae, under the family Blennocampidae.

Lacourt [56] classified the Tenthredinidae into 14 subfamilies, including Athaliinae, which lies between Selandriinae and Emphytinae.

Based on morphological characters, Vilhelmsen [29] reconstructed the phylogeny of the extant early-diverging lineages of Hymenoptera. Within the superfamily Tenthredinoidea, *Athalia* was located between the Blaticotomidae and other taxa of the superfamily (Figure 1b).

Schulmeister [30] revised the data matrix of Vilhelmsen [29], provided additional morphological data, and reconstructed the phylogeny of early-diverging Hymenoptera. This study demonstrated that the systematic position of the genus *Athalia* was uncertain, with one possible position being the first lineage of Tenthredinoidea except for Blasticotomidae (Figure 1c).

Since Schulmeister [31], *Athalia* has repeatedly been placed outside of the Tenthredinidae (Figure 1d), even though almost all of these studies were based on reduced taxon sampling (e.g., Figure 1e) [32,57,58]. In the studies of Boevé et al. (Figure 1f) [34] and Malm and Nyman (Figure 1g) [33] with more exhaustive taxon sampling, the genus *Athalia* was placed in an early-diverging position within the Tenthredinidae or as a rogue lineage in the tree due to the Heptamelinae, both poorly supported.

Vilhelmsen et al. [59] analyzed the phylogeny of apocritan wasps based on the structure of mesosoma. Four sawfly taxa were included in the study, and the results demonstrated that the relationship of the four taxa was (*Notofenusa* + (*Athalia* + (*Monoctenus* + *Heteroperryia*) or (*Athalia* + (*Monoctenus* + (*Notofenusa* + *Heteroperryia*).

Vilhelmsen [60] performed the most recent morphological phylogenetic analysis of Tenthredinidae. Although the generic and subfamilial relationships of Tenthredinidae were poorly resolved, his results indicated that *Athalia* was not a ‘normal member’ of Allantinae.

The results of these previous studies are of particular interest because they suggested that *Athalia* was possibly a distinct basal lineage of Tenthredinoidea (excluding Blasticotomidae). However, their scarce sampling, namely the lack of closely related genera such as *Hypsathalia* or *Dentathalia,* and limited molecular data (i.e., short DNA sequences), is not sufficient to support the monophyly of an “*Athalia*” clade. Such studies have led to confusion in tenthredinid classification, with some researchers placing them into Allantinae (Tenthredinidae). For example, Taeger et al. [61] compiled a world catalog of Symphyta and divided the Tenthredinidae into six subfamilies, and placed *Athalia*, *Hypsathalia*, *Hennedyia,* and *Hennedyella* into the Allantinae.

The present study aims to clarify the position of the genus *Athalia* and its relatives, for which we constructed a comprehensive time-calibrated phylogeny of Tenthredinoidea based on both mitochondrial and nuclear single-copy genes and by also increasing taxon coverage by including *Hypsathalia* and *Dentathalia*. We assessed the reliability of their taxonomic status by considering the evolutionary history and by investigating mt-genomic features among the “Symphyta”.

## 2. Materials and Methods

### 2.1. Taxon Sampling

The genome and/or mitogenome data of 80 sawfly species were used in this study (Table 1). The sampling includes 76 tenthredinoid species representing 53 genera from seven families and four xyeloid species as outgroups. The genome and mitogenome data of 23 and only mitogenome data of seven out of these species were sequenced and reported in this study. The specimens were provided by Jiangxi Normal University, China. The data of the remaining 44 species were retrieved from NCBI database (Table 1). From these species, RNA/DNA-seq FASTQ files of 30 species were downloaded from the GenBank SRA database, while the mitogenome sequences of 14 species were retrieved from the GenBank Nucleotide database.

### 2.2. DNA Extraction and Sequencing

Total genomic DNA was extracted from the thorax muscle of each ethanol-preserved specimen by the DNeasy Blood & Tissue Kit (Qiagen, Valencia, CA, USA) according to the manufacturer’s instructions, and the extracted DNA samples were quantified using a NanoDrop (Maestrogen, Inc., Waltham, MA, USA). The genomic DNA extracts were then pooled and subjected to 150-bp paired-end reads on the Illumina HiSeq 4000 platform (sequenced by Shanghai Majorbio Bio-pharm Technology Co., Ltd., Shanghai, China). Approximately 10 Gb raw data were generated for each library.

### 2.3. Genome Assembly and Single-Copy Assignment

The genome assemblies of NGS reads generated in this study and previously deposited to SRA database were performed using the pipeline plws v1.0.6 (https://github.com/xtmtd/PLWS, accessed on 1 October 2021) suggested by [91]. The reads were firstly grouped into clumps and duplicates were removed with clumpify.sh (BBTools suite version 38.91; [92]). The quality controls and normalizations were performed with bbduk.sh and bbnorm.sh under BBTools. The Minia v3.2.6 [93] was used for contig assembly with multiple k-mers and the REDUNDANS v0.13a5 [94] was preferred for detection and selective removal of redundant contigs. Scaffolding of the assembled contigs and gap-filling were performed with BESST v2.2.8 [95] and GAPCLOSER v1.12 [96], respectively. The scaffolds were then searched against hymenoptera_odb10.2020-08-05 database composed of 5991 genes using Benchmarking Universal Single-Copy Orthologs BUSCO v5.2.2 [97] to obtain the Single-copy Orthologs (SCOs) of each species. Orthologous sequences were combined in separate fasta files as amino acid sequences, removing duplicated and fragmentary genes.

### 2.4. Mitogenome Assembly, Annotation, and Structure Predictions

To construct the mitogenome assemblies of the species, the NGS reads were trimmed with BBDuk, and then the high-quality reads were assembled de novo using MIRA assembler implemented in Geneious R11 (Biomatters Ltd., Auckland, New Zealand) [98] The obtained assemblies were mapped using the reference symphytan mitogenomes (accession numbers: NC024664, NC045360, NC057102, NC056796, MG923517) under ‘medium-low sensitivity’ parameters for each species. The annotations of the mitogenomes were carried out using MITOS2 (http://mitos2.bioinf.uni-leipzig.de/index.py, accessed on 29 January 2021) [99]. The boundaries and locations of the protein-coding genes (PCGs) and rRNA genes were manually checked by comparing the reported symphytan homologous gene sequences. Considering the complexity of gene rearrangements in Hymenoptera, we performed the validation of genes and adjacent regions involved in gene rearrangements. For the same reason, the obtained reliable GOs were not used to reconstruct the ancestral state, but rather to compare the GO states for observing the fit between GO features and the phylogeny obtained by sequence-based methods. The procedure of rRNA secondary structure prediction is referred to in the previous study [100]. Instead of focusing on the variability within a taxon, this study is more concerned with the stability within family-level units and the inter-family variability, in other words, exploring potentially shared derivatives for different groups to obtain support for the taxon rank.

### 2.5. Phylogenomic Analyses

#### 2.5.1. Alignment, Refinement, Supermatrix Construction, and Model Selection

Each fasta file including SCOs was aligned with MAFFT v7.471 [101] using the algorithm L-INS-i. Poorly and ambiguously aligned regions within the amino acid alignments were removed with Gblocks v0.91b [102] using default parameters except for setting the options ‘minimum number of sequences for a flank position’ to 15 and ‘minimum length of a block’ to 4. The supermatrix was then produced by concatenating the individual alignment files using concat-aln (https://github.com/aberer/concat-aln, accessed on 1 November 2021). The optimal partitioning scheme and the best-fitting substitution models were estimated by PartitionFinder v2.1.1 [103] for the SCOs dataset including the parameters of the among-site variation (+G) and parameters of amino acid frequency estimation (+F). The substitution models were further restricted to 11 amino-acid substitution models (LG + G, WAG + G, DCMUT + G, JTT + G, BLOSUM62 + G, LG + G + F, WAG + G + F, DCMUT + G + F, JTT + G + F, BLOSUM62 + G + F, LG4X) because these are the most estimated models for the insect studies [1,2].

The alignment of each PCGs of the mitochondrial gene was performed with the MAFFT algorithm implemented in Geneious R11, preferring the ‘translation align’ option. The rRNA was performed with MAFFT using the algorithm E-INS-i. The resulting alignments were concatenated using PhyloSuite [104], and the PCG, PCG + RNA, as well as AA datasets, were generated. Given the compositional heterogeneity across the datasets including PCG, the CAT-GTR model in PhyloBayes [105] was used. The cross-validation analyses were performed for AA datasets under the homogeneous models (MtArt) and heterogeneous models (CAT + GTR).

#### 2.5.2. Phylogenomic Inferences

The phylogenomic reconstruction of Tenthredinoidea based on the supermatrix of SCOs was carried out using the maximum likelihood (ML) and Bayesian inference (BI) approaches. Two xyeloid species were selected as outgroups: *Xyela alpigena* (Strobl, 1895) and *Megaxyela euchroma* Blank, Shinohara and Wei, 2017. The ML analysis was performed in IQ-TREE v2.1.4 [106] using default parameters under the Q.plant + R10 (-lnL: 14280658.547) evolutionary model inferred by PartitionFinder v2.1.1. Node support was estimated with 10000 bootstrap replicates using the fast bootstrapping option implemented in IQTree. The BI analysis was carried out using ExaBayes v1.5.1 [107] under default parameters with automatic substitution model detection. Four coupled Markov chain Monte Carlo (MCMC) chains were run with one million generations by sampling every 500 generations. To check whether the chains had achieved stationarity, ‘burn-in’ plots were evaluated by plotting log-likelihood scores and tree lengths against generation numbers using the software Tracer v1.6 [108]. After assessing for convergence, the first 20% of trees were discarded as burn-in and a majority-rule consensus tree (BI tree) was generated using Geneious R11 from the remaining trees. Visualization of the trees was performed by FigTree v1.4.2 [109].

For the mitochondrial datasets, Bayesian analyses were run with Phylobayes-MPI 1.9 [105]. For all analyses, two runs were performed and convergence was investigated using the bpcomp option. The analysis was stopped when the conditions considered to indicate a good run were reached (maxdiff < 0.1) or sufficient effective sample size was reached (effsize > 300).

#### 2.5.3. Divergence Time Estimation

Divergence times of the tenthredinoid lineages were estimated using the supermatrices of SCOs and mitochondrial genes by MCMCTree program implemented in the Phylogenetic Analysis by Maximum Likelihood (PAML) package v4.9 [110]. Three fossil calibration points were used: (i) *Triassoxyela foveolata* Rasnitsyn, 1964 for stem calibration of Xyelidae, (ii) *Pseudoxyelocerus bascharagensis* Nel, Petrulevicius and Henrotay, 2004 for stem calibration of Tenthredinoidea and (iii) *Abrotoxyela lepida* Gao, Ren and Shih, 2009 for the crown calibration of Xyelidae. Two additional molecular calibration points were set as (i) 168.5 Ma for the split between Pergidae and Argidae [111], and (ii) 73 Ma for the crown age of Cimbicidae [26]. The oldest estimated age for holometabolous insects [112] was preferred for the constraint of the root (<400 Ma). Substitution rate per site for the supermatrix of SCOs was estimated by CODEML and was used to set the prior for the mean substitution rate in the MCMCTree. The MCMC was run by 50 × 10,000 iterations with the JTT substitution model [113]. The convergence was assessed by considering effective sample sizes (ESS > 200) using Tracer v1.7 [108] and the maximum clade credibility tree was generated by TreeAnnotator v1.10.4 after removing 25% of the trees as burn-in. The trees produced by both datasets were visualized in FigTree v1.4.2 [109].

## 3. Results

### 3.1. Phylogenomic Assessment of the Placement of Athalia and Its Relatives

#### 3.1.1. Phylogenetic Analysis Based on Sequences

The mitochondrial DNA sequence (Mt) and single-copy orthologs (SCOs), representing two different markers, yielded consistent interfamily relationships. The monophyly and the position of the ‘*Athalia*’ clade were supported with high support values across all analyses (Figure 2, Figure 3 and Figure 4). When using Xyelidae for rooting, the Blasticotomidae were an earlier-diverging lineage. The tree then split into two branches, one of which consisted of Argidae and Pergidae, with a predominantly Southern Hemisphere distribution. The other branch consisted of families distributed mainly in the Northern Hemisphere, with a basal ‘*Athalia*’ clade. Diprionidae and Cimbicidae were supported as sister clades, forming the closest relatives of Tenthredinidae *s. str.* All of the above clades were recovered as monophyletic with high posterior probabilities (PPs, >0.90) except for the Pergidae, which has only one representative, so monophyly of this family cannot be ensured.

The extant Athaliidae are distinct from Tenthredinidae, including Heptamelinae, and supported by the following morphological characteristics: 1. Anepimeron largely and roundly convex with only a small basin above (apomorphic; the basic state of anepimeron is flat, as shown in Xyelidae); 2. Pleural suture distinctly curved in the upper third (apomorphic); 3. Specialized penis valve (apomorphic); 4. Prepectus large and isolated (plesiomorphic); 5. vein 1 M meeting vein Rs instead of vein R (plesiomorphic; reversed in some genera of Tenthredinidae); and 6. Long and filiform antenna with more than 10 antennomeres (plesiomorphic; reversed in some genera of Tenthredinidae). The convex anepimeron is also present in the family Argidae, but it is convex overall and without a small basin above. In Cimbicidae, the anepimeron is generally more or less concave, but in several genera, the lower end of the anepimeron is distinctly convex. In Diprionidae, the whole anepimeron is elevated to form a platform.

Unlike the consistent high-level relationships, the relationships between subfamilies or equivalent levels varied with datasets and models. The SCO under ML analysis and PCG datasets supported the Heptamelinae as the early diverged group of Tenthredinidae; nevertheless, the support for this position was low (PP = 0.54) in PCGs (Figure 3a) and median in SCOs (Figure 4a, 0.86). In the analyses of AAs under both models (Figure 2) and of PCGs + RNA (Figure 3b), the Heptamelinae were recovered as the sister of (Cimbicidae + Diprionidae), with support for this position ranging from 0.59 to 0.92, and in the BI analyses of SCOs (Figure 4b), it was supported with placement at the base of (Tenthredinidae + (Cimbicidae + Diprionidae)). Whether it should be separated from the Tenthredinidae as an independent family needs a full reassessment. Another inconsistency occurs in Selandriinae and Nematinae. They are consistently located at the base of Tenthredinidae (excluding Heptamelinae); however, they are successively split out in two AA (Figure 2, PP > 0.81) and SCOs analyses (Figure 4) and exhibit a sister group relationship in the PCG-RNA (Figure 3b, PP = 0.51) analysis or form an unresolved triple branch with other species in the PCG (Figure 3a, PP = 0.99) analysis.

At least four subclades can be recognized within the ‘*Athalia’* clade. In addition to *Dentathalia* and *Hypsathalia*, *A.* ‘tibetana’ morphotype() and *A. tanaoserrula* Chu & Wang, 1962 formed an independent subclade. All three cases of the relative positions of the latter two subclades were recovered, with both being sister groups to each other (Figure 3b: PCG-RNA, PP = 0.89) or diverging successively but in a different order. The subclade of *A.* ‘tibetana’ and *A. tanaoserrula* at the basal position was supported in SCOs (Figure 4), as well as AA-CAT-GTR (Figure 2a, PP > 0.72) and PCG (Figure 3a, PP > 0.98). This suggests that the diversity within the ‘*Athalia’* clade has not yet been fully captured.

#### 3.1.2. Rare Genomic Changes

To compare rare gene changes, we mapped GOs and secondary structure onto the phylogenetic backbone obtained from the PCG dataset (Figure 5).

In the ‘*Athalia’* clade, *trnC* and *trnW* had moved upstream relative to those in Xyelidae, leaving a single *trnY* at the *nad2-cox1* junction. In addition, the shuffling within the trn-IMQ cluster was also an autapomorphy. While *trnT* and *trnP* were variable across the ‘*Athalia’* clade, they swapped positions at least twice. If the GO of the Xyelidae is representative of the ancestral state of Hymenoptera, then gene rearrangement seems to have occurred in two directions in the subsequent divergence. One is the minor rearrangement represented by Blasticotomidae, and the other is rather major, which involved the tRNA crossing of *nad2*.

The base composition and predicted secondary structures are conserved across the diverse taxa in the ‘*Athalia*’ clade (Appendix A). In *rrnS*, domain III was more conserved within the ‘*Athalia’* clade. In particular, helix H939 was completely conserved among the twelve samples. However, there were only three known patterns: the shared pattern of Xyelidae and Orussidae; the pattern of Cephidae, Siricidae, and Pamphilioidea; and the pattern of the Tenthredinoidea including Blasticotomidae. Compensatory base-pair changes (CBCs) have taken place in the second and fifth pairs. The phylogenetic signals carried by this helix seem to be more suitable for defining higher-level diversity. In *rrnL,* helices H533, H563, H579, H736, H822, H1775, H1830, H2023, H2259, and H2064 are completely conserved. However, only some of them can be treated as signals at multiple hierarchies.

### 3.2. Divergence Time Estimates

There was a slight variance between the topologically constrained analyses, broadly overlapping 95% credible intervals for most nodes (Figure 6). Two inferences consistently restrict several early divergences, including the origin of Athaliidae, to the pre-Cretaceous. For Athaliidae specifically, these included 186.86 Ma (95% HPD = 167~206 Ma) inferred from mtG analysis or 168.33 Ma (95% HPD = 162~174 Ma) based on the SCO dataset. The estimates for the age of crown Athaliidae differed strongly, depending on the datasets. However, both results fell within the Cretaceous, ranging from 133.18 Ma (95% HPD = 109~156 Ma) inferred from mtG analysis to 143.46 Ma (95% HPD = 127~171 Ma) based on the SCOs dataset. The phase of species diversification could not be captured due to the scarcity of species-level samples.

## 4. Discussion

### 4.1. Sequence-Based Phylogeny, Rare Genomic Changes, and Morphology: Congruence in the Placement of Athaliidae

Resistance to the independence of Athaliidae from Tenthredinidae *s. str.* comes from its morphological similarity to Allantinae [61]. Our results indicate that such subjective equivalencies can mask profound evolutionary differences, as argued by Sterelny [114]. A more detailed morphological comparison reveals that the wing venation pattern originally thought to be shared by the Allantinae and *Athalia* is a plesiomorphic character, which does not support their monophyly, but they even have different origins and represent different evolutionary pathways. The wing venation of the Athaliidae has a consistent hint of curvature, which is a continuation of an ancient wing pattern of Hymenoptera. In contrast, the wing venation of Allantinae does not have such a bend and tends to be simplified, a derived feature in Hymenoptera [115,116].

Fossils provide complementary evidence for the placement of Athaliidae. *Palaeathalia laiyangensis* Zhang, 1985, discovered in eastern China, resembles extant *Hennedyella* by sharing numerous plesiomorphic characters, such as antenna long and slender with more than 13 antennomeres, cell 1 M broader with a short dorsal petiole, and the vein cu-a meeting cell 1 M at the middle. The main differences between *Palaeathalia* and *Hennedyella* are that cell C of the forewing in *Palaeathalia* is very broad, and the head is strongly dilated behind the eyes in dorsal view. Within the extant taxa of Athaliidae, only *Hypsathalia* shares a very broad cell C in the forewing. If we regard *Palaeathalia laiyangensis* as an ancient fossil representative of the former Allantinae, it pushes the divergence of this subfamily to at least 127 Ma. However, a long gap exists between it and the earliest confirmed record of the now-dominant Tenthredinidae in the early Paleocene (*Tenthredo primordialis* Piton, 1940, 61 Ma) [117]. If Athaliidae is located outside of Tenthredinidae and (Cimibicidae + Diprionidae), as this study reveals, then it would make sense for this clade to be represented by ancient fossils.

This speculation has received support from molecular analyses [25,31,33] and some morphological studies [28,55] demonstrating that *Athalia* and its relatives differ from the Allantinae. Uncertainties in their position come from the following two issues: (1) the relative positions of the Athaliidae and Heptamelinae and, (2) the rank of the monophyletic clade including *Athalia* and its relatives. In other words, whether the Athaliidae have a large enough gap with others to match the family rank. Phylogenomics provides the confidence needed to choose among existing hypotheses and sheds new light on genome evolution.

Sequence-based phylogenetic analyses of both mt genome data and SCOs genome data support the monophyletic Athaliidae standing between (Argidae + Pergidae) and (Cimbicidae + Diprionidae). Beyond the linear sequence comparisons, the genome-level characters also support this relationship (Figure 5). Athaliidae and all other families have their own unique ground patterns of GOs, which not only are incompatible between families but also allow the inference of reasonable GOs evolution scenarios in a broader evolutionary context (Figure 5). The GO of Heptamelinae is more similar to that of the Tenthredinidae. If it stands in place of present Athaliidae, that is, between (Argidae + Pergidae) and (Cimbicidae + Diprionidae) [33], it indicates either parallel evolution in Heptamelinae or reversal to an ancestral GO pattern in Athaliidae.

Structural inference is difficult and tedious and hence avoided in the overwhelming majority of published studies [118]. However, the intensive sampling here reduces the uncertainty in this process. The predicted structures (Appendix A) are highly conserved, while the mutation sites exhibit robust phylogenetic signals. Compared with *rrnS*, the pairing on the stem of *rrnL* shown in Figure 5 has strong lineage specificity. These discrete characters imply evolutionary independence above the species level. It could be argued that the mt-genomic distinctness of *Athalia* and its relatives warrant recognition at the rank of family, as could be claimed for all of the well-supported primary lineages within Tenthredinoidea.

In summary, the Athaliidae should be recognized as an isolated family that falls outside of Tenthredinidae, Cimbicidae, and Diprionidae. We suggest that the Tenthredinoidea, therefore, be considered an evolutionary clade consisting of six well-defined families and a ‘core Tenthredinidae’. Among them, three Neotropical/Southern Hemisphere families, Argidae, Pergidae, and Zenargidae, form a clade sister to the predominantly East Asian taxa, which include three small families, Cimbicidae, Diprionidae, and Athaliidae, and the ‘core Tenthredinidae’.

### 4.2. Spatial and Temporal Diversification of Athaliidae

The morphology of the clypeus is a crucial trait in understanding the classification of *Athalia* [39]. The backbone in this study corresponds in a general way to clypeus evolution. For example, *A.* ‘tibetana’ and *A. tanaoserrula* Chu and Wang, 1962 correspond to the ‘*A. cordata* group’ in having a subsquare clypeus, while *A. icar* Saini and Vasu, 1997 and all other samples of *Athalia* form a monophyletic lineage corresponding to the ‘*A. rosae* group’ [39] in having a short and weakly convexed clypeus. However, morphological studies indicate that each clypeus type is present in species from two or more distribution types. This study on the phylogeography of Athaliidae reveals that the ‘*A. vollenhoveni* group’ (Benson, 1962; clypeus narrowly incised) may be the sister group of the ‘*A. roase* group’, while the ‘*A. himantopus* group’ (clypeus broadly and very shallowly emarginated) is possibly related to the ‘*A. bicolor* group’ and ‘*A. cordata* group’, and Benson’s ‘*rosae* group’ (clypeus short with anterior margin weakly convex) is probably not monophyletic (Niu et al., in preparation). Revision of the family still requires the addition of representative samples from African, Indian, and European taxa.

Malaise [60] discussed the biogeography of *Athalia* in detail and its absence from America. According to his theory, *Athalia* was relatively ancient and already present early in the Cenozoic. This hypothesis was later supported by the description of fossil species of the genus from the Eocene by Florissant [119]. Benson [39] discussed the distribution of Athaliini. He pointed out that the two genera *Hennedyia* and *Hennedyella* were known only from the western Mediterranean and Burma, respectively, and were probably from the early Cenozoic. The high Himalayan genus *Hypsathalia* was likely derived from lowland Cenozoic stock, possibly the same stock that gave rise to some species of *Athalia*. Benson [39] also discussed the distribution of characters and the possible origin of some species groups of *Athalia* from Europe and Asia. The origin of the diversified species of *Athalia* in the Afrotropical region was uncertain in his opinion.

Our analyses suggest that the most recent common ancestor (MRCA) for the crown Athaliidae lived in the Late Cretaceous. This coincides with the origin of their primary hosts [120]. However, the Athaliidae originated before the Cretaceous (186 Ma), yet, the Brassicales and Lamiales are both attributed to the later clades in the Cretaceous radiation of the core eudicots (90–87 Ma). In other words, the origin of the hosts of the extant species emerged later than the stem group of Athaliidae. It can be speculated that the MRCA of Athaliidae may have undergone host transfer during angiosperm radiation in the Cretaceous. This may explain the uniqueness of the Athaliidae in terms of the host. Araceae and *Sedum* L., 1753 were previously reported as hosts for the Eurasian species. They represent Alismatales and Saxifragales, which originated approximately 150 Ma and 126 Ma, respectively. They may be candidates for the ancestral host plants of Athaliidae.

When considering that the origin of the Brassicaceae [121] and Lamiales [122] occurred nearly simultaneously at both the order level (90~87 Ma) and the family level (35 Ma), it is more likely that the early divergence of Athaliidae underwent at least two completely different evolutionary trajectories, accompanying the rapid rise of the major clades of angiosperms. This of course cannot exclude the possibility of host shifts between Lamiales and Brassicaceae [123].

Most of the East Asian taxa of Athaliidae exhibit a distinctly plain, low-mountain distribution type, feeding on cultivated crucifers. However, some branches may be adapted to high-altitude habitats and are more likely to feed on Lamiales. Clades occurring on the Qinghai–Tibetan Plateau (QTP) and in adjacent areas show phylogenetic diversity in either alpine grassland (central and western Tibet) or alpine meadows (Qinghai, northeastern Tibet). Based on a few observations during field collection, it is likely that the representative dominant taxa of the plateau, *Sedum*, are their new hosts.

Unlike the several Eurasian widespread basal branches of the Athaliidae, three other ancient species, *Hennedyia annulitarsis* Cameron, 1891, *Hennedyella athaloides* Forsius, 1935 and *Hennedyella typica* Saini and Vasu, 1996, are distributed in Gibraltar and the Eastern Himalayas. A plausible explanation for the distribution of ancient species at the edge of the diversity center is competition with other successful radiating clades. When new branches acquire ‘key innovations’ such as gene rearrangements, they likely have wider niches or richer diversity, while older branches are likely to suffer.

Such a case is prominent in the tRNA cluster upstream of *nad2*; for the first time in Hymenoptera, *trnC* was transposed upstream of *nad2* from the WCY cluster downstream of *nad2*. Specifically, the tRNA clusters showed the pattern CWMQI in *Dentathalia* and CWQMI in the rest of Athaliidae. Considering the ground pattern of the members within the sister clade of Athaliidae (Cimbicidae: MQI; Tenthredinidae: MQI), it is presumed that the GO of *Dentathalia* represents the primitive type. Given the uniqueness of its genome structure and several primitive morphological characters, we propose that the basal lineage, *Dentathalia*, be recognized at the genus level. In addition, *A.* ‘tibetana’ and *A. tanaoserrula* also have a unique TP shift, and such an event has been reported only for the Diprionidae. Its phylogenetic signal and role in the mitogenome structure evolution of Hymenoptera still need to be further evaluated. However, the rearrangement was also accompanied by specificity in the CG content of the flanking PCGs. The possibility that this clade should be considered an independent genus cannot be ruled out, and it is therefore recommended that the genus-level status of *Hypsathalia* should be retained (with at least four undescribed species distributed on the QTP), as being supported by placement between (*A.* ‘tibetana’ + *A. tanaoserrula*) and *Dentathalia*.

Despite being able to confidently reestablish the genus-level status of *Dentathlia* and retain the genus-level status of *Hypthathalia*, this work was not able to provide sufficient resolution to understand the tribal classification within Athaliidae. Although some major lineages could be recognized, the absence of African species has hindered the further study of evolution and diversification patterns within and among clades. Thus, it is too early to propose a formal tribal classification of this family.

Several issues remain to be addressed. The inference that the reversals would be a small-probability event when complexity is taken into account satisfies one of the principal criteria for a more accurate phylogenetic marker [15]. However, this inference was at high risk of bias for the Hymenoptera. For example, *trnTP* rearrangements occur in two distant taxa, one *Athalia* clade, and the Diprionidae. On the other hand, gene rearrangements appear to be a key innovation that triggers diversification in Athaliidae; however, not all clades demonstrate the independent pattern. Therefore, we doubt that gene rearrangement itself was a direct driver of increased generic diversification. It can also be reasoned that the direct observation of rearrangement mechanisms acting on microevolution in genus-level diversification may be a very small-probability event. However, this does not prevent GO, as a feature of multiple hierarchical levels, from being considered as evidence supporting macroevolutionary results. The coexistence of parallelism and variability among proxies for low-level clades as well as at the family level in Hymenoptera suggests that the application of GO changes needs more exploration.

This evolutionary clade of the Athaliidae does not seem to exhibit key morphological innovations compared to the next bursts of Tenthredinidae, which is the reason why it was proposed to stay within the family. However, when gene sequences and genomic structures were observed, we were able to speculate that some traits originated long before. That is, there is a macroevolution lag [124] between the origin of the Athaliidae and the crown Athaliidae and the subsequent dramatic diversity of the Tenthredinidae. Considering the probable existence of feeding exploration in Athaliidae and their morphological similarity to the Allantiniae, we have to consider the possibility that, as a result, the Athaliidae appear to be similar to a clade of Tenthredinidae in a preadapted manner. In other words, at the origin of the Athaliidae, the Hymenoptera appeared intrinsically ready to undergo diversification until an external factor provided a ‘critical opportunity’ [125].

## 5. Conclusions

The present study comprehensively assessed the position of *Athalia* and its relatives using sequence-based methods and genomic characterization. Inferences under the phylogenomic framework revealed a monophyletic Athaliidae positioned between Blasticotomidae and the rest of Tenthredinoidea. The mitogenome characters provided strong phylogenetic information to support the independence of the respective families. The present results call for a new classification of Tenthredinoidea. The diversification within the Athaliidae dated back to at least the upper Cretaceous, with the second phase of divergence occurring shortly after the K-Pg extinction. It is reasonable to speculate that the diverse fauna has a Eurasian origin, while the African fauna is secondary and may have multiple origins. With the uplift of the Himalayas, at least two clades adapted to the cold and settled at a 3000-m altitude, probably favoring *Sedum*. Most East Asian species stayed in the plain region and colonized a variety of Brassicaceae. However, only one plateau branch has gene rearrangements and a specific base composition of the corresponding genes, which calls for denser sampling to clarify the intergeneric relationships of *Athalia*. In addition, increasing data from African and European species for phylogeographic studies will be essential to understanding the Eurasian–African distribution pattern of extant insects.

## Figures and Tables

**Figure 1 insects-13-00858-f001:**
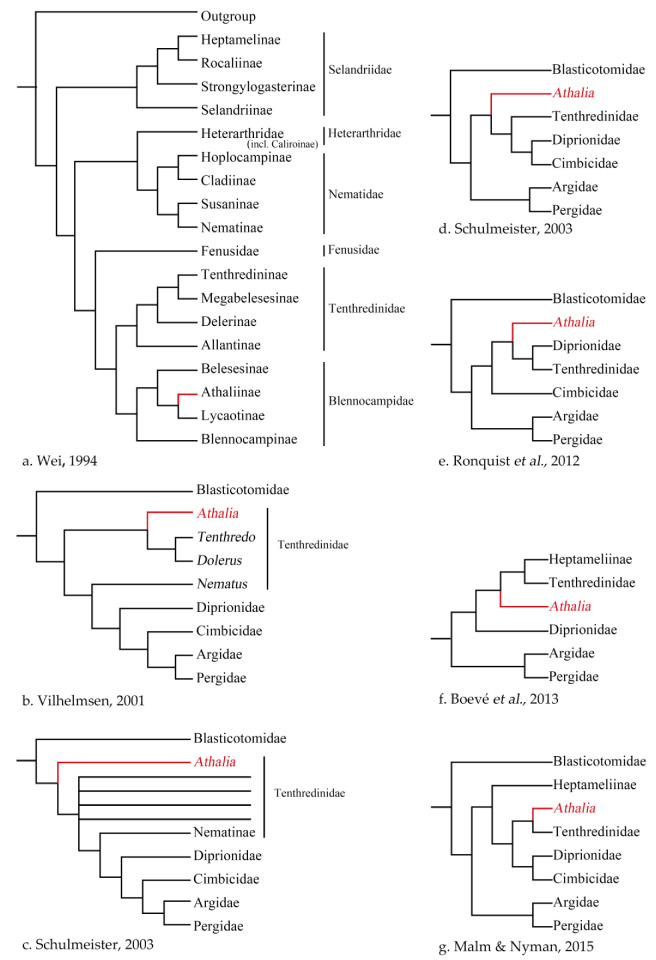
Schematic phylogeny for basal hymenopteran lineages simplified from the reference provided below each cladogram. (**a**) Wei, 1994 [28], Figure 5-1, morphological data; (**b**) Vilhelmsen, 2001 [29], Figure 11, morphological data; (**c**) Schulmeister, 2003 [30], Figure 7, morphological data; (**d**) Schulmeister, 2003 [31], Figure 3, morphological and molecular data; (**e**) Ronquist et al., 2012 [32], Figure 3, morphological and molecular data; (**f**) Malm and Nyman, 2015 [33], Figure 4, molecular data; (**g**) Boevé et al. 2013 [34], Figure 3, molecular data.

**Figure 2 insects-13-00858-f002:**
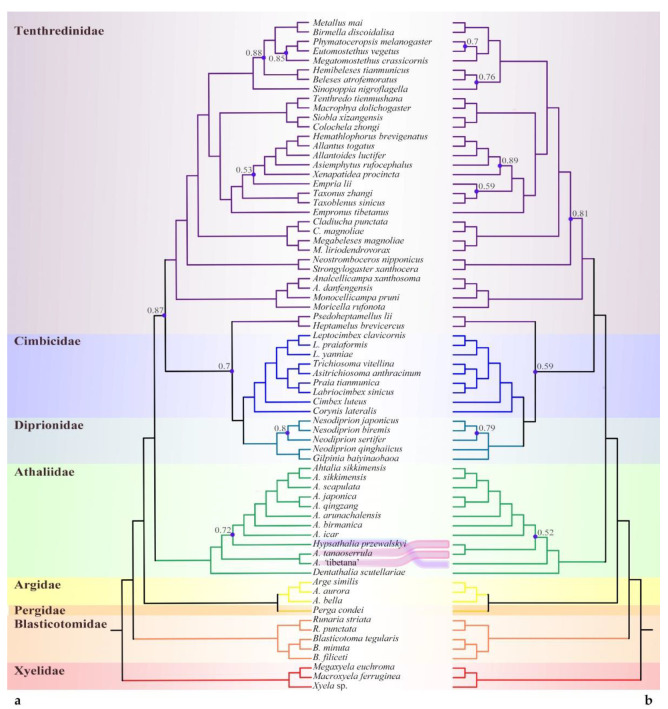
Topological comparison of PhyloBayes trees based on the concatenated amino acid matrix under (**a**) the CAT-GTR model and (**b**) the MtArt model, showing species group sampling and points of agreement and conflict. Posterior probability support values less than 0.90 are displayed on the trees with blue dots.

**Figure 3 insects-13-00858-f003:**
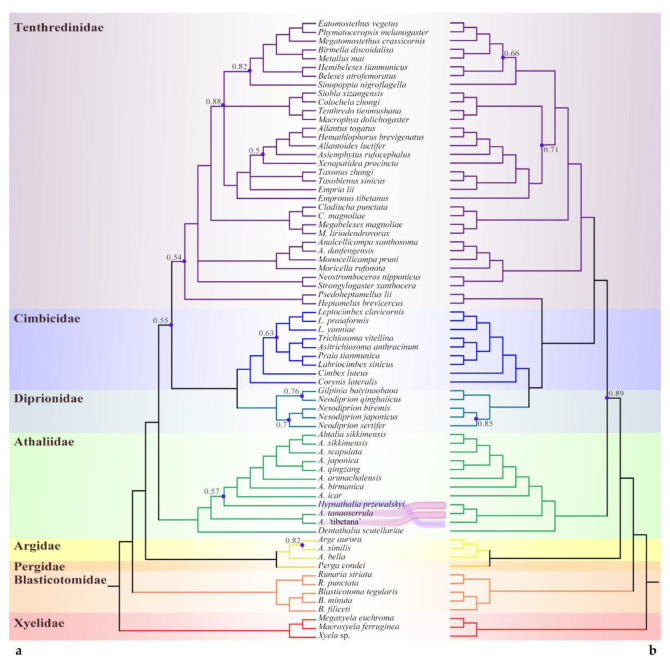
Topological comparison of PhyloBayes trees based on the concatenated nucleotide sequences of (**a**) the 13 PCGs and (**b**) the 13 PCGs and 2 rRNAs, showing species group sampling and points of agreement and conflict. Posterior probability support values less than 0.90 are displayed on the trees with blue dots.

**Figure 4 insects-13-00858-f004:**
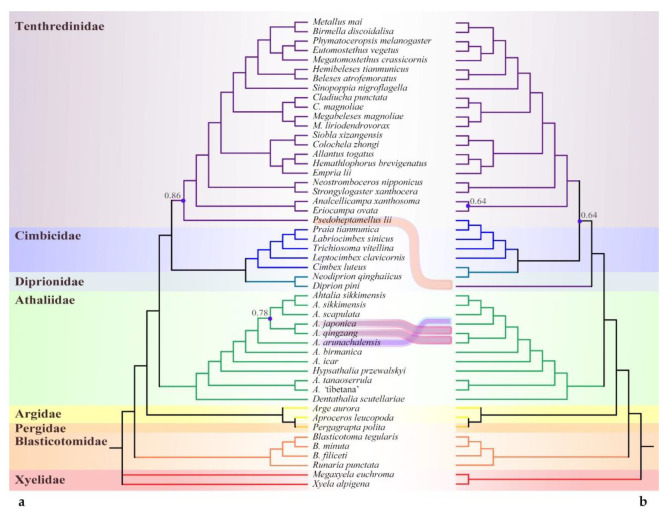
Topological comparison of trees based on single-copy orthologs (SCOs). (**a**) Iqtree; (**b**) PhyloBayes. Bootstrap values and posterior probability support values less than 0.90 are displayed on the trees with blue dots.

**Figure 5 insects-13-00858-f005:**
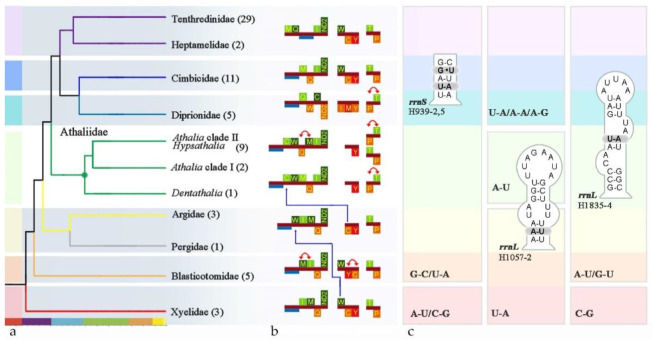
High-level taxon tree, gene rearrangement, and autapomorphic secondary structures of rRNA. (**a**) The dendrogram was simplified from the PCG dataset under the CAT-GTR model; numbers in brackets represent the mitogenome used in phylogenetic inference; (**b**) outline of rearrangement events across the Tenthredinoidea, with the blue lines indicating a major translocation and the red lines indicating a minor translocation; (**c**) group-specific structural characters of mt *rrnL*.

**Figure 6 insects-13-00858-f006:**
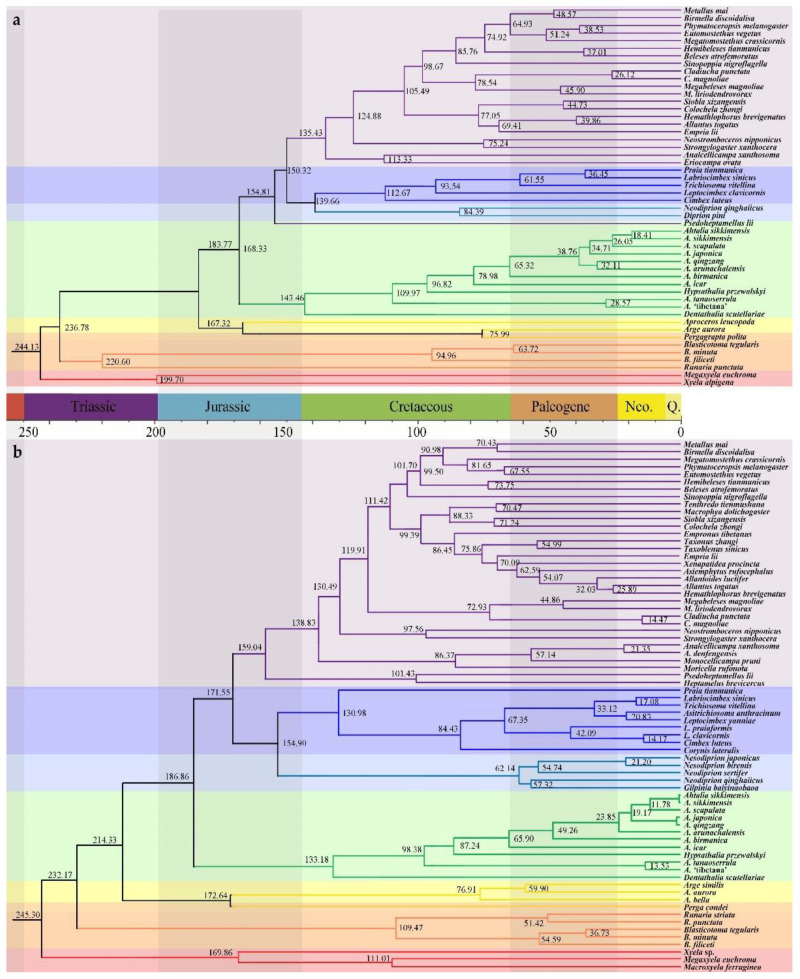
Age estimation trees using the dataset of (**a**) mtDNA; and (**b**) SCOs, highlighting the early diversity of Athaliidae constrained to the pre-Cretaceous.

**Table 1 insects-13-00858-t001:** Summary information of mitogenomes and the Single-copy Orthologs (SCOs) used in phylogenetic analyses.

Family	Species	Accession Number	References	BUSCO	Matrix	Voucher Number
Blasticotomidae	*Runaria striata*	ON964462	This study	/	/	CSCS-Hym-MC0014
	*Runaria punctata*	ON808427	This study	5321	4877	CSCS-Hym-MC0019
	*Blasticotoma minuta*	ON964461	This study	5576	5115	CSCS-Hym-MC0060
	*Blasticotoma tegularis*	ON840089	This study	5484	5037	CSCS-Hym-MC0083
	*Blasticotoma filiceti*	ON840091	This study	5343	4901	CSCS-Hym-MC0140
Pergidae	*Perga condei*	AY787816	[62]	/	/	/
	*Pergagrapta polita*	/	SRX642961	2292	2132	/
Argidae	*Arge bella*	MF287761	[63]	/	/	CSCS-Hym-MC0008
	*Arge similis*	MG923484	[64]	/	/	/
	*Arge aurora*	MN913350	This study	5298	4864	CSCS-Hym-MC0179
	*Aproceros leucopoda*	/	SRX642913	3615	3378	/
Athaliidae	*Athalia icar*	MN527306	[65]	2184	1934	CSCS-Hym-MC0021
	*Athalia birmanica*	ON840085	This study	5789	5312	CSCS-Hym-MC0077
	*Athalia japonica*	ON964466	This study	5713	5251	CSCS-Hym-MC0081
	*Athalia sikkimensis*	ON840087	This study	5672	5219	CSCS-Hym-MC0345
	*Athalia scapulata*	ON840088	This study	5662	5193	CSCS-Hym-MC0080
	*Athalia arunachalensis*	ON840086	This study	5794	5320	CSCS-Hym-MC0355
	*Athalia rosae*	/	ERX8976896	4480	4165	/
	*Athalia* ‘tibetana’	ON964467	This study	/	/	CSCS-Hym-MC0731
	*Athalia qingzang*	ON964468	This study	/	/	CSCS-Hym-MC0732
	*Athalia tanaoserrula*	ON964469	This study	/	/	CSCS-Hym-MC0733
	*Athalia sikkimensis*	ON964470	This study	/	/	CSCS-Hym-MC0734
	*Hypsathalia przewalskyi*	ON840090	This study	5701	5240	CSCS-Hym-MC0187
	*Dentathalia scutellariae*	ON808426	This study	5780	5310	CSCS-Hym-MC0360
Cimbicidae	*Cimbex luteus*	MW136447	[66]	607	502	CSCS-Hym-MC0035
	*Labriocimbex sinicus*	MH136623	[67]	5338	4897	CSCS-Hym-MC0009
	*Leptocimbex clavicornis*	MT478109	[68]	5423	4973	CSCS-Hym-MC0162
	*Leptocimbex praiaformis*	MT478110	[68]	/	/	CSCS-Hym-MC0167
	*Leptocimbex yanniae*	MT478111	[68]	/	/	CSCS-Hym-MC0133
	*Praia tianmunica*	MT665975	[69]	4092	3710	CSCS-Hym-MC0049
	*Asitrichiosoma anthracinum*	KT921411	[70]	/	/	/
	*Trichiosoma vitellina*	MN853777	[71]	5186	4743	CSCS-Hym-MC0165
	*Corynis lateralis*	KY063728	[72]	/	/	/
Diprionidae	*Neodiprion sertifer*	MK994526	[73]	/	/	/
	*Neodiprion qinghaiicus*	ON964471	This study	5657	5192	CSCS-Hym-MC0198
	*Gilpinia baiyinaobaoa*	ON840092	This study	/	/	CSCS-Hym-MC0178
	*Nesodiprion biremis*	ON964465	This study	/	/	CSCS-Hym-MC0055
	*Nesodiprion japonicus*	ON964464	This study	/	/	CSCS-Hym-MC0010
	*Diprion pini*	/	SRX642914	3933	3651	/
Heptamelinae	*Psedoheptamellus lii*	ON964463	This study	5653	5202	CSCS-Hym-MC0175
	*Heptamelus brevicercus*	MW632128	[25]	/	/	CSCS-Hym-MC0018
Tenthredinidae	*Allantoides luctifer*	KJ713152	[74]	/	/	/
	*Allantus togatus*	MW464859	[75]	5676	5206	CSCS-Hym-MC0142
	*Asiemphytus rufocephalus*	KR703582	[70]	/	/	/
	*Empria lii*	MW632124	[25]	4737	4323	CSCS-Hym-MC0079
	*Empronus tibetanus*	MZ265343	[25]	5148	4703	CSCS-Hym-MC0152
	*Hemathlophorus brevigenatus*	MW632125	[76]	4789	4373	CSCS-Hym-MC0177
	*Taxoblenus sinicus*	MW632126	[77]	/	/	CSCS-Hym-MC0193
	*Xenapatidea procincta*	MW487928	[25]	/	/	CSCS-Hym-MC0022
	*Taxonus zhangi*	MZ461490	[78]	/	/	CSCS-Hym-MC0342
	*Beleses atrofemoratus*	MZ265347	[25]	1520	1311	CSCS-Hym-MC0011
	*Eutomostethus vegetus*	MT663219	[79]	5398	4939	CSCS-Hym-MC0184
	*Megatomostethus crassicornis*	MZ265345	[25]	5682	5222	CSCS-Hym-MC0075
	*Phymatoceropsis melanogaster*	MZ265346	[25]	947	792	CSCS-Hym-MC0032
	*Monophadnus latus*	*/*	SRX6352475	4237	3914	/
	*Birmella discoidalisa*	MF197548	[80]	494	399	CSCS-Hym-MC0029
	*Metallus mai*	MW255941	[25]	5643	5168	CSCS-Hym-MC0182
	*Sinopoppia nigroflagella*	MW487927	[81]	5405	4945	CSCS-Hym-MC0073
	*Colochela zhongi*	MT702984	[82]	5714	5246	CSCS-Hym-MC0061
	*Macrophya dolichogaster*	MW544890	[25]	/	/	/
	*Siobla xizangensis*	MN562486	[83]	5753	5280	CSCS-Hym-MC0150
	*Eriocampa ovata*	/	[84]	5811	5337	CSCS-Hym-MC0143
	*Tenthredo tienmushana*	KR703581	[85]	/	/	/
	*Tenthredo koehleri*	*/*	SRX314899	4563	4230	/
	*Cladiucha magnoliae*	MT295305	[25]	2104	1844	CSCS-Hym-MC0015
	*Cladiucha punctata*	MT295306	[25]	5733	5266	CSCS-Hym-MC0044
	*Megabeleses liriodendrovorax*	MW255939	[25]	1808	1570	CSCS-Hym-MC0030
	*Megabeleses magnoliae*	MW255940	[25]	5793	5316	CSCS-Hym-MC0042
	*Neostromboceros nipponicus*	MW632127	[25]	817	679	CSCS-Hym-MC0026
	*Strongylogaster xanthocera*	MW324676	[86]	5727	5255	CSCS-Hym-MC0070
	*Analcellicampa xanthosoma*	MH992752	[25]	1250	1066	CSCS-Hym-MC0017
	*Analcellicampa danfengensis*	MN163004	[87]	/	/	/
	*Monocellicampa pruni*	JX566509	[88]	/	/	/
	*Moricella rufonota*	MW487926	[89]	/	/	CSCS-Hym-MC0068
	*Nematus ribesii*	*/*	SRX643001	3544	3329	/
	*Hemibeleses tianmunicus*	MZ265344	[25]	5072	4631	CSCS-Hym-MC0027
Xyelidae	*Xyela* sp.	MG923517	[64]	/	/	/
	*Xyela alpigena*	/	SRX642930	3517	3308	/
	*Megaxyela euchroma*	OL794667	This study	5393	4938	CSCS-Hym-MC0001
	*Macroxyela ferruginea*	MK270536	[90]	/	/	/

## Data Availability

The data presented in this study are openly available in Figshare at https://figshare.com/account/home#/projects/141053, accessed on 8 June 2022.

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
