# Peer review of "Phylogenomic Analyses of the Tenthredinoidea Support the Familial Rank of Athaliidae (Insecta, Tenthredinoidea)"

_insects, 2022, doi:10.3390/insects13100858_

Round 1

Reviewer 1 Report

I am very happy to see the authors adress the controversial issue of the classification of the Athalia lineage with an ambitious analysis, that adds substantial data for the solution. Methodology seems sound and empirical work solid.

However, the presentation of the question in the introduction is unnecessary long and detailed and provides the entire history of classification of the group, which is nice for the interested few but should in a research paper preferrably be limited to summarising the background and the different attempts at dealing with the specific topic at hand.

Also, the results and the suggested taxonomic changes are unnecessarily strongly phrased. This is a significant new contribution and a comprehensive analysis, but it still possible that other analyses may yield different results and this is by no means a final solution. Thus, taxonomic rearrangements are preferrably phrased as suggestions based on the results, and they may be followed or not by subsequent researchers.

Both these problems become accentuated by the insufficient English of the manuscript. It is necessary to have a fluent English-speaker with  experience with phylogenetic work to thoroughly edit this manuscript. Actual errors in grammar and spelling are quite few throughout the manuscript, but the phrasing and style is mostly awkward, using unusual words, non-idiomatic phrases, convoluted rhetorics, and causing what seems to maybe be statements of simple facts to be very difficult to interpret, and/or with logical errors arising. This concerns also the phylogenetic terminology and the taxonomic terminology, where opaque or unusual phrasing seems to lead to the authors making several strange and probably unintended statements.

Thus, the actual empirical content of the manuscript seems to be of high quality and well worth publishing, but it needs to be entirely rewritten in a style that matches this high quality.

Author Response

On behalf of all the contributing authors, I would like to express our sincere appreciation for your review and the recognition of the quality of the content.

We have also noticed that you mentioned awkward phrasing and style, unusual words, and other problems. In particular, this bad writing leads the reader to believe that we overemphasize results. Therefore, we used the language editing service from <Springer Nature Author Services>. The manuscript was edited for grammar, phrasing, and punctuation. In addition, many edits were made to further improve the flow and readability of the text. Apart from the language editing service, we got feedback from several colleagues and made some improvements to the manuscript.

We deleted a lot of details in the introduction as you suggested. What remains of the research history is the result of a cumbersome taxonomy. To avoid the repetition of this time-costing taxonomic study by future researchers, we try to solve this problem once and for all in this manuscript.

Thank you again for your valuable comments to improve the quality of our manuscript.

Reviewer 2 Report

The article is sufficiently novel and interesting to warrant publication for the reason that it defines a new family, Athaliidae apart from the family Tenthredinidae. The latter was made upon a comprehensive phylogenomic study of Tenthredinidae, focused on the position of Athalia and its related genera by sampling 71 representatives of Tenthredinoidea and phylogenetic reconstructions based on nuclear genes and mitochondrial sequences. It does add to the canon of knowledge and adheres to the standards of the journal « Insects » because of the comparison of symphytan mitochondrial genomes demonstrating the innovative gene rearrangement pattern of Athalia and closely related genera.

The article is structured in the right way, including 1. Introduction; 2.Material and methods; 3. Results; 4. Discussion; 5. Conclusions; Supplementary material consists of two figures and a Reference list of 132sources quoted in the main text. The « Introduction » summarizes the relevant research and provides context, explaining what other authors’ findings are being challenged or extended. It is too lengthy (row range 43 to 267) and could be shortened without the danger of weakening the value of the publication. For instance, Opitz et al. (quoted to reference title number 67) had their results and conclusions far apart from the theme of the present investigation. The « Material and methods » include Taxon sampling, DNA extraction and sequencing, Genome assembly and single-copy assignment, Mitogenome assembly, annotation, and structure predictions, Phylogenomic analyses, Phylogenomic inferences, and Divergence time estimation. The authors accurately explain how the data was collected and sampling is appropriate; the equipment and materials had been adequately described. The design is suitable for answering the posed question. The article identifies the followed procedures in detail and these latter are ordered in a meaningful way. The « Results » including Phylogenomic assesses the placement of Athalia and its relatives and Divergence time estimates are clearly laid out in a logical sequence and well-illustrated. The « Discussion » including Sequence-based phylogeny, rare genomic changes, and morphology: congruence in the placement of Athaliidae and Spatial and temporal diversification of Athaliidae explains in modest view how the research has moved the body of scientific knowledge forward. One exception here is worth denoting: line 570-587: I am not convinced of the “Brassicales and Lamialesare primary hosts” for the new family Athaliidae (as supported by any previousresearch) and the whole two packs of sentences here seems rather speculative and I propose just to be removed from the MS.

Author Response

On behalf of all the contributing authors, I would like to express our sincere appreciation for your careful review.

We deleted a lot of details in the introduction as you suggested. What remains of the research history is the result of a cumbersome taxonomy. To avoid the repetition of this time-costing taxonomic study by future researchers, we try to solve this problem once and for all in this manuscript.

The conclusion concerned Opitz et al. has also been deleted from the introduction. Given your feedback, we realize that the mention of this sentence reinforces the reader's impression that hosts are primarily Brassicales and Lamialesare. And that's not what we want. Nor do we agree that Brassicales and Lamialesare are the primary hosts. Because in terms of Brassicaceae and Lamiales, both at the order level (90~87 Ma) or at the family level (35 Ma), they are too young for Athaliidae (186 Ma). However, we are sorry that our unskilled writing has caused unnecessary misunderstanding. We have adjusted the wording to emphasize host diversity.

Considering that these misunderstandings may come from language, we used a language service from <Springer Nature Author Services>. The manuscript was edited for grammar, phrasing, and punctuation. In addition, many edits were made to further improve the flow and readability of the text. Apart from the language editing service, we got feedback from several colleagues and made some improvements to the manuscript.

Thank you again for your valuable comments to improve the quality of our manuscript.